# A Critical Appraisal of the Measurement of Adaptive Social Communication Behaviors in the Behavioral Intervention Context

**DOI:** 10.3390/bs15060722

**Published:** 2025-05-23

**Authors:** Thomas W. Frazier, Eric A. Youngstrom, Allison R. Frazier, Mirko Uljarevic

**Affiliations:** 1Department of Psychology, John Carroll University, 1 John Carroll Blvd, University Heights, OH 44118, USA; 2Departments of Pediatrics and Psychiatry, SUNY Upstate Medical University, Syracuse, NY 13210, USA; 3Autism Speaks, Princeton, NJ 08540, USA; 4Institute for Mental and Behavioral Health Research and the Department of Psychiatry, Nationwide Children’s Hospital and The Ohio State University, Columbus, OH 43215, USA; eric.youngstrom@nationwidechildrens.org; 5Elevate Learning, LLC, Pepper Pike, OH 44124, USA; allison@ohelevate.com; 6Department of Psychiatry and Behavioral Sciences, Stanford University, Stanford, CA 94305, USA; mirkoulj@stanford.edu

**Keywords:** autism spectrum disorder, behavioral intervention, Vineland-3, ABAS-3, adaptive behavior

## Abstract

Despite encouraging evidence for the efficacy of comprehensive and intensive behavioral intervention (CIBI) programs, the majority of studies have focused on relatively narrow, deficit-focused outcomes. More specifically, although adaptive social communication and interaction (SCI) are essential for facilitative functioning, the majority of studies have utilized instruments that capture only the severity of SCI symptoms. Thus, given the importance of the comprehensive and appropriate characterization of distinct SCI adaptive skills in CIBI, in this review, based on PubMed search strategies to identify relevant published articles, we provide a critical appraisal of two of the most commonly used adaptive functioning measures—the Vineland Adaptive Behavior Scales-Third Edition (Vineland-3) and the Adaptive Behavior Assessment System-Third Edition (ABAS-3), for characterizing SCI in the behavioral intervention context. The review focused on periodic outcome and treatment planning assessment in people with autism spectrum disorder receiving CIBI programs. Instrument technical manuals were reviewed and a PubMed search was used to identify published manuscripts, with relevance to Vineland-3 and ABAS-3 development, psychometric properties, or measure interpretation. Instrument analysis begins by introducing the roles of periodic outcome assessment for CIBI programs. Next, the Vineland-3 and ABAS-3 are evaluated in terms of their development processes, psychometric characteristics, and the practical aspects of their implementation. Examination of psychometric evidence for each measure demonstrated that the evidence for several key psychometric characteristics is either unavailable or suggests less-than-desirable properties. Evaluation of practical considerations for implementation revealed weaknesses in ongoing intervention monitoring and clinical decision support. The Vineland-3 and ABAS-3 have significant strengths for cross-sectional outpatient mental health assessment, particularly as related to the identification of intellectual disability, but also substantial weaknesses relevant to their application in CIBI outcome assessment. Alternative approaches are offered, including adopting measures specifically developed for the CIBI context.

## 1. Introduction

Over the last several decades, a range of targeted practices that use the principles and practices of behaviorism and more comprehensive intervention packages that typically integrate multiple specific practices with an over-arching discipline or philosophy (e.g., applied behavior analysis or developmental psychology) have been developed and evaluated in youth with autism spectrum disorder (ASD) ([32]; [67]). Despite encouraging evidence for the efficacy of the noted approaches, in particular, with regard to improvements in IQ and language abilities, the majority of treatment trials have focused on a relatively narrow range of outcomes, emphasizing the need to develop assessments capturing behaviors and skills that are relevant to a person’s ability to function across different daily contexts. Further, over the last decade, both theoretical frameworks and dimensional initiatives, such as the National Institute of Mental Health’s Research Criteria, and latent variable modeling studies, have emphasized the multi-dimensional nature of specific aspects of adaptive functioning, in particular, the social communication and interaction domain, demonstrating that different subdomains have distinct trajectories and underpinning mechanisms and might thus respond differently to specific treatments. Therefore, by providing only overly broad scores that conflate distinct subdomains, outcome measures can obscure the potential positive effects of specific treatments. In addition to the importance of domain coverage and representation, modern psychometric approaches have made significant strides towards regression-based norming and other approaches that can be utilized to optimize instruments to inform treatment selection as well as to quantify treatment-related change at the individual patient level and thus significantly improve the science and practice of behavioral interventions. Crucially, with the recognition of the importance of neurodiversity-affirming practices, it has become increasingly clear that it is essential to go beyond simply focusing on symptom-/deficit-based outcomes and approaches ([57]; [64]; [74]). Given these recent developments, it is essential to ensure that outcome measures are psychometrically robust and provide a comprehensive capture of all the relevant adaptive subdomains without conflating distinct constructs. Further, it is important to establish whether currently used instruments have utilized state-of-the-art psychometrics to provide modern norming, the ability to track change, and facilitate practice through automated online administration, scoring, and interpretation. Thus, in this review, we focused on appraising whether two of the most widely used adaptive functioning instruments—the Vineland Adaptive Behavior Scales-Third Edition (Vineland-3) and the Adaptive Behavior Assessment System-Third Edition (ABAS-3), provide construct coverage in line with the current empirical evidence and incorporate psychometric advances to enable their valid use as outcome measures for behavioral treatments.

### 1.1. Types of CIBI Packages

The most common comprehensive and intensive behavioral intervention (CIBI) packages for ASD can be grouped into a more structured or traditional application of applied behavior analysis (ABA) that includes early intensive behavioral intervention approaches that utilize a more directive and structured approach to intervention ([8]) and naturalistic developmental behavioral interventions (NDBIs) that emphasize following the child’s lead, using naturalistic reinforcement in everyday settings to promote developmentally appropriate skills ([54]). Importantly, even in research applications of ABA and NDBI, clinicians often modify the application of the intervention package to the characteristics of the child, potentially even borrowing from features of the other package in implementation ([52]). For instance, in situations where the need for stimulus and environmental control is high, naturalistic delivery is less crucial or relevant, and/or child-led intervention is less likely to be effective, many ABA practitioners, including Board Certified Behavior Analysts, need to adapt their intervention delivery methods to the specific profile of individual’s characteristics, using aspects of ABA with strong developmental consideration. In addition, it is essential for practitioners to consider how meaningful and valid specific treatment goals are to the given family and individual ([64]; [74]). Regardless of individual perspectives on the ABA and NDBI approaches, a growing number of individuals are accessing CIBI packages, and, as a result, there is a need for useful and appropriate outcome measures to inform the process and evaluate the value of these treatments and to consider a reduction in specific symptoms and, equally importantly, the development of specific aspects of adaptive functioning and quality of life ([74]).

### 1.2. Assessment of Social Communication/Interaction Skills in CIBI

Although a major focus of CIBI has, historically, been on reducing the severity of social communication and interaction (SCI) symptoms, it has also been suggested that it is crucial for treatments to focus on the development and improvements in the specific SCI adaptive skills that underpin one’s ability to navigate the complexity of the social world, and, if impaired, result in the symptoms that characterize ASD. However, one of the major issues in terms of evaluating the effects of the specific treatments has been the fact that the most widely used instruments often conflate skills and symptoms, and adaptive SCI behaviors are often the other side of the autism symptom coin, with SCI skills being replacement behaviors for core autism symptoms. The highly similar nature of SCI measurement, when conceptualized as an adaptive skill or as a symptom, is apparent when comparing social communication and interaction items across adaptive functioning and autism symptom measures. For instance, the following item stem used to measure the socialization domain on the Vineland-3 “Realizes when others are happy, sad, …” is highly similar to the following item measuring perspective taking on the Autism Symptom Dimensions Questionnaire “Seem to understand what others are thinking or feeling”. Strong item similarity is not an exception, as many adaptive behavior items assessing SCI behaviors overlap significantly with social communication content on autism symptom measures (see Appendix A).

Given the prominence of SCI behaviors as intervention targets for individuals with ASD, including their important contribution to more positive long-term outcomes ([19])), the present paper focuses on the potential of adaptive function measures as measures of SCI within CIBI outcome assessment instruments. Specifically, we first briefly review historical and research approaches to evaluating the benefits and outcomes of CIBI. Then, the potential roles of and needs for outcome assessment in CIBI are covered. Next, modern conceptualizations of SCI behavior are reviewed to set the stage for evaluation of two of the most commonly administered adaptive functioning measures, the Vineland Adaptive Behavior Scales-Third Edition [Vineland-3] ([63]) and Adaptive Behavioral Assessment System [ABAS-3] ([30]), as measures of SCI behavior. Evaluation of these instruments as measures of SCI for CIBI includes a review of how the measures were developed, current evidence supporting their development, psychometric characteristics (including factor structure, measurement invariance, reliability, and validity), and practical aspects of their implementation to inform CIBI practice. Finally, this evaluative review ends by discussing how the field can shift toward more optimized approaches to adaptive behavior outcome assessment as part of a larger strategy for assessing CIBI outcomes and informing care for people with ASD.

### 1.3. Historical Approaches to Outcome Assessment for CIBI

As different treatment models, including the NDBI and ABA models, were adopted by specialty care centers and payor coverage became more widespread, the need for intermittent assessment of cognition and behavior as a means of evaluating and demonstrating the value of the everyday community implementation of CIBI increased ([60]). The batteries of measures implemented often depended on the specific treatment model or philosophy, including a mixture of standardized and unstandardized and proximal (e.g., measurement of behaviors as they occur during treatment) versus distal (e.g., assessment of cognitive or behavioral constructs at given timepoints during the intervention course) measurement strategies. In general, periodic outcome assessments frequently included instruments assessing general cognitive ability, language, speech/communication, adaptive function, and autism symptoms ([51]). A standardized, robust, well-validated, comprehensive set of assessments capturing all aspects of the clinical phenotype as well as other key cognitive and adaptive skills is essential to ensure that every patient is assigned to appropriate treatments and that their responses to interventions are sensitively monitored to enable changes in clinical management. However, substantial variability has been present within and across clinical and research contexts. This variability has even extended across payor policies for behavioral intervention delivery models. More specifically, while some payors are adopting specific measure sets that are required to be collected at regular intervals (often every 6–12 months), such as TRICARE (https://www.tricare.mil/autism (accessed on 1 August 2024)), other payors and trade organizations are providing general guidance for measure selection or leaning on existing best practices and recommendations (https://www.casproviders.org/standards-and-guidelines (accessed on 1 August 2024)). Within the last 18 months, organizations have even begun to recommend autism-specific outcome batteries ([34]).

### 1.4. Current Challenges in CIBI Outcome Assessment

Given the current state of the assessment field and the sometimes poorly specified or capricious nature of requirements or recommendations within and across funder policies, several initiatives have attempted to address these issues, with significant impact on both clinical practice and policy. For example, the National Academies of Science Engineering and Medicine (NASEM) was asked by Congress to evaluate the Autism Care Demonstration (ACD) project within the TRICARE military health benefit. Part of the ongoing analysis being performed by NASEM is expected to include making recommendations regarding future outcome assessment strategies (https://www.nationalacademies.org/our-work/independent-analysis-of-department-of-defenses-comprehensive-autism-care-demonstration-program (accessed on 1 August 2024)). Through the process adopted by NASEM, specific concerns have been raised about whether TRICARE ACD legacy assessment approaches are appropriate for outcome assessment of CIBI, whether measures are covering meaningful domains of functioning relevant to patients and families, and practical concerns regarding the feasibility of measure completion and whether measures are being appropriately scored and utilized to inform intervention.

Thus, there is a crucial need as well as an opportunity to develop and validate more rational, evidence-based, or evidence-informed outcome assessment practices for CIBI in ASD. This need is occurring against a backdrop of significant advances in psychological assessment, including the increasing adoption of recommendations for measure development practices ([4]; [18]) as well as technical advances in how measures are psychometrically evaluated and practically delivered. More specifically, over the last decade, there have been substantial improvements in the evaluation of measure structure ([40]), norming ([62]), scoring for longitudinal monitoring ([16]), assessment of reliable and meaningful change ([5]), and practical improvements in instrument deployment such as automated online administration, scoring, reporting, monitoring, and the provision of decision support ([7]). Unfortunately, the process of the development, validation, and deployment of the most widely used outcome assessment of CIBI has not incorporated noted psychometric advances that are essential for delivering instruments fully optimized for comprehensive characterization, score interpretation, and treatment monitoring.

In light of the current circumstances, and given the nascent nature and growing availability of CIBI for ASD and the fact that a substantial proportion of all BCBAs are less than 5 years from certification (https://www.bacb.com/bacb-certificant-data/ (accessed on 1 August 2024)), there is opportunity and motivation to re-think CIBI outcome assessment at the instrument and process levels. This is particularly crucial in light of recent discussions in the field emphasizing the need to incorporate perspectives from the community and focus on outcomes identified by autistic individuals and their families as being important, meaningful, and socially valid ([3]; [55]; [56]). Thus, this review focuses on understanding the characteristics and potential utility of existing adaptive function measures within CIBI outcome assessment, with an eye toward what improvements might be possible with additional advances in measure development, evaluation, and implementation.

### 1.5. Multiple Roles for Assessment Processes Within CIBI

In 2024, the Council of Autism Service Providers published The Applied Behavior Analysis Practice Guidelines for the Treatment of Autism Spectrum Disorder Guidance for Healthcare Funders, Regulatory Bodies, Service Providers, and Consumers ([9]), which includes a description of initial and periodic assessment activities as well as considerations for monitoring and adjusting treatment in order to achieve optimal outcomes. After a diagnosis of ASD, and a referral for CIBI services, generally accepted standards of care include using a multi-modal assessment approach to inform the CIBI treatment plan and ongoing clinical management (Appendix A). CIBI assessment methods can be broadly categorized into three types: (1) session-based data collection on specific intervention targets relevant to pre-specified objectives, (2) the collection of intervention quality and fidelity data, and (3) periodic outcome and treatment planning assessment. All three assessment types inform the provision of services, including addressing below-expectation skills and reducing or replacing maladaptive behaviors. Each assessment type adds distinct value to CIBI, but the present review focuses on periodic CIBI outcome and treatment planning assessments. For additional information on session-based and fidelity/quality assessment see Appendix A.

A rigorous, periodic (often every 6 months) CIBI outcome and treatment planning assessment process (Appendix A) is often narrowly conceptualized as determining the strength of the individual’s intervention response at a given point in time, enabling stakeholders (including payors or health plans) to identify patient progress and inform future clinical management, including authorizations of services for additional treatment periods. While this is an important aspect of CIBI outcome assessment, it is not the only value that the application of outcome measures can provide. Instead, and consistent with existing guidelines ([9]), this review posits at least five major roles that initial and ongoing CIBI outcome assessment can fulfill: (1) developing the initial CIBI strategy, (2) identifying cognitive and behavioral domains and sub-domains, (3) recommending specific behaviors to address within treatment protocols, (4) monitoring intervention progress and determining the strength of intervention response, and (5) highlighting possible intervention modifications and changes in clinical management. It is crucial to consider these roles when evaluating the Vineland-3 and ABAS-3 as measures of SCI adaptive functioning, and to understand how future measurement instruments, including revisions of these instruments, could provide additional value. In addition, it is important to consider the recent criticisms of the exclusive deficit-focused orientation of the more traditional CIBI approaches and the lack of alignment between traditionally defined intervention outcomes and the outcomes that were identified as most meaningful by autistic individuals and their families, including outcomes that are related to improvements in different aspects of quality of life ([3]; [64]). Appendix A provides detailed information about specific roles for periodic CIBI outcome and treatment planning assessments.

### 1.6. Modern Understanding of SCI Structure and Relevant Assessment Considerations

There have been substantial developments in the understanding of the structure and taxonomy of specific facets of adaptive social communication and interaction behavior. For instance, both theoretical frameworks ([29]; [31]) and single- and multi-instrument factorial work ([6]; [22]; [48]; [59]; [70]) have highlighted the multifaceted nature of social and communication functioning, identifying a specific set of processes underpinning one’s ability to adapt to the complexities of daily life that distinct and often non-linear development progression. Specifically, detailed, large-sample factor-analytic work using neurotypical, ASD, and other neurodevelopmental disability samples has identified an overarching social communication and interaction behavior domain (in contrast to separate domains for social interaction and communication) and at least four subdomains of social communication and interaction behavior (see Appendix A). The identified sub-domains include social motivation or affiliation-related behaviors, basic social communication behaviors (including verbal and non-verbal aspects), perspective-taking or theory of mind-related behaviors (comprising both cognitive and emotional aspects), and the frequency and quality of relationship-focused and reciprocal interaction behaviors. Importantly, the noted domains and sub-domains emerged across the factor analyses of both instruments that capture SCI abilities/functioning (e.g., Stanford Social Dimensions Scale ([48]) and SCI-related symptoms (e.g., Social Responsiveness Scale, Social Communication Questionnaire, and Autism Diagnostic Interview-Revised) ([25], [26], [24], [20], [21]; [22]; [27]; [38]; [58]; [61]; [70], [71]). These domains are also aligned with current transdiagnostic frameworks, including the National Institute of Mental Health’s Research Domain Criteria ([33]).

Given the complex and multifaceted nature of the SCI domain, it is crucial that CIBI outcome assessment batteries include SCI domain coverage and representation that is aligned with noted conceptual and empirical advancements to ensure the adequate measurement of the full breadth of SCI behaviors. If specific SCI dimensions are not explicitly assessed or are only partially measured, it will be difficult to build and monitor an appropriate intervention plan. For example, in a child with lower social motivation and weaker basic social communication skills but with a recently developed friend, clinicians may be tempted to focus heavily on perspective-taking and relationship skills but would be better served by addressing the more basic social motivation and basic social communication skills before moving toward more complex and higher-level social cognitive skills. The principle of strong sub-domain coverage has particular relevance for adaptive function measures, which were not developed with core ASD symptom dimensions or a modern understanding of SCI behavior and, therefore, tend to focus on the very broad measurement of the SCI construct. Reinforcing the importance of capturing distinct aspects of SCI are recent studies that have demonstrated the utility of fine-grained SCI subdomains for characterizing ASD heterogeneity and identifying informative subgroups. For instance, a relatively recent study utilizing a comprehensive SCI measure has identified five subgroups of youth with ASD that showed distinct patterns of strengths and weaknesses across different aspects of the social motivation, basic social communication skill, and theory of mind SCI subdomains rather than simply reflecting a severity gradient and, importantly, the identified subgroups were further differentiated in terms of cognitive ability, the severity of ASD symptoms, and the severity of co-occurring internalizing symptoms ([72]).

Evidence-based assessment frameworks point out that the clinical goals of diagnosis and treatment planning versus progress, process, and outcome evaluation not only occur at different phases in the arc of a course of treatment, but they also often have distinct features that can sometimes be in tension ([79]). Given that adaptive function instruments were not specifically developed with CIBI in mind, but are nevertheless widely deployed in the CIBI outcome assessment process, it is essential to examine how these measures might inform specific clinical activities in the context of the necessary roles fulfilled by the periodic CIBI outcome assessment process (Appendix A). Additionally, as part of evaluating the Vineland-3 and ABAS-3 in terms of their ability to measure SCI behavior and fulfill the above-described roles within periodic CIBI outcome assessment, it is useful to consider their development processes and, in particular, the lack of alignment with the modern conceptual and empirical understanding of the SCI domain, psychometric characteristics, and practical considerations in their deployment. Thus, before providing a detailed evaluation of these instruments, we first outline the measure development process and the psychometric and practical features needed for any instrument contributing to effective CIBI outcome measurement. Then, within each section, we focus on evaluating these considerations for the measurement of SCI by the Vineland-3 and ABAS-3.

## 2. Review Methods and Results

For the present critical review, two literature searches were conducted using PubMed. For the Vineland-3, search terms included Vineland AND adaptive AND (psychometric OR reliability OR validity OR development). For the ABAS-3, search terms included ABAS OR adaptive behavior assessment system AND adaptive AND (psychometric OR reliability OR validity OR development). The PubMed searches generated 749 results for the Vineland and 1822 results for the ABAS. The titles and abstracts were reviewed for relevance, and the reference lists were reviewed for articles with strong relevance to identify any missed publications. The Vineland-3 and ABAS-3 technical manuals were also reviewed for relevant information.

Over the last 15 years, several efforts have been made to provide guidance to measure developers regarding best practices in instrument development. In particular, the FDA has provided substantial guidance regarding patient-reported or observer-reported outcome measure development ([18]). Similarly, measure development experts have recommended sets of procedures and processes that can be used to inform measure development and subsequent psychometric evaluation and validation ([4]). Table 1 presents the measure development guidelines, based on expert recommendations, that are most relevant to application in CIBI and their relationship to Vineland-3 and ABAS-3 development. These guidelines were used to evaluate the Vineland-3 and ABAS-3 development and validation processes. Appendix A provide additional information on the literature search.

### 2.1. Measure Development Processes

Both the Vineland-3 ([63]) and ABAS-3 ([30]) include stakeholders with developmental disabilities in the measure development process and report in their manuals methods which appear to have involved evaluation of item characteristics, including relevance, clarity/readability, and potential biased language. However, neither instrument’s manual description includes mention of parent/caregiver or patient/client stakeholder inclusion at the outset of conceptualization, using concept elicitation or cognitive interviews, or creating a domain map with stakeholders. From the manuals, it does not appear that much of the development work occurred within a behavioral intervention context, as the primary initial development rationale emphasized the measurement of adaptive behavior as part of the identification of intellectual and developmental disability (IDD). Furthermore, it is apparent that the full developmental disability population was not considered, as, for example, the Vineland-3 sub-domains and items (e.g., Communication sub-domain: Reading and Writing) have very limited relevance to those with significant cognitive challenges or profound autism ([37]). As a result, neither the Vineland-3 nor ABAS-3 were purpose-built for the CIBI outcome assessment context.

### 2.2. Psychometric Considerations

The psychometric properties most important to CIBI outcome assessment are listed in Table 2, including an overview of existing evidence for the Vineland-3 and ABAS-3 relevant to each characteristic. While a comprehensive psychometric evaluation of any psychological assessment instrument can involve a wide range of methods and characteristics, the present review focuses on those most crucial for the assessment of social communication and interaction behavior within a CIBI context.

#### 2.2.1. Overview of Instrument Structure and Scoring

The Vineland-3 ([63]) is scored with up to 4 domain scores, including separate socialization and communication domains. The socialization and communication domains include 6 subscale scores (Socialization: Interpersonal Relationships, Play and Leisure, Coping Skills; Communication: Receptive, Expressive, Written). A 3-point Likert frequency scale is used for each item that corresponds to “Never”, “Sometimes”, and “Usually/Often”. The ABAS-3 ([30]) has two broad domains and three skill areas relevant to SCI behavior (Conceptual: Communication and Social: Social and Leisure). The items use a 4-point Likert frequency scale—“Never”, “Sometimes”, “Often”, and “Very Often”. In addition to the parent/caregiver report versions, the Vineland-3 offers an interview form and both the Vineland-3 and ABAS-3 offer teacher report versions. The teacher report versions are less relevant to a CIBI context where additional raters are often not available or obtaining reliable information from them is not feasible. The interview version is recommended in the Vineland-3 manual ([63]), with the implication that introducing clinical judgment yields more accurate or valid measurement. However, this review did not identify any peer-reviewed publications that specifically demonstrate greater validity for the interview version of the Vineland-3 (see Appendix A for additional information). While clinical judgment has a longstanding and well-worn history of being used to integrate information and clarify diagnostic and phenotypic presentations, there are numerous examples where clinical judgment actually introduces substantial variability across interviewers or sites ([49]), a significant problem for real-world CIBI programs where staff turnover can be frequent. While it is certainly possible that clinical judgment does enhance accuracy or validity, the absence of evidence, particularly with regard to the specific contexts in which clinical judgment might provide additional insights, raises significant concerns as to whether the significant cost, clinician time, and burden for families related to the use of the interview form of the Vineland-3 are justified. Further, this represents a major challenge for some clinics where providers may not have sufficient training in clinical interviewing.

#### 2.2.2. Factor Structure and Measurement Invariance

Although often overlooked, before considering measure scoring, it is critical that the structure of the instrument be understood. Recent developments in latent structural analysis permit the detailed comparison of competing models, with best-fitting models being used to inform instrument scoring. Unfortunately, the factor-analytic literature of the Vineland-3 and ABAS-3 is very limited, both in terms of the number of structural evaluations but also regarding the use of a limited range of methods for identifying optimal structure. Of the published factor-analytic evaluations of the Vineland-3 ([10]; [17]; [46]; [76]), the analyses used only subscales as indicators (not items) and the results were inconsistent, suggesting from one to three factors but often with poor or adequate but not excellent model fit. For example, two of the identified Vineland-3 confirmatory factor-analytic studies reported poor fit ([46]; [76]), below the recommended levels for adequate or excellent comparative fit indices ([39]). Furthermore, measurement invariance and differential item functioning analyses suggested potential measurement inconsistencies across IDD and other non-IDD populations ([41]), although structural consistency was found for sex, race/ethnicity, and SES ([63]). The factor structure literature for the ABAS is even more sparse, with only two published studies, possibly using the same dataset, examining the US, Taiwanese, and Romanian versions and suggesting support for the three major domain factors across versions with evidence of the gender-invariant measurement of the proposed scoring model ([44]; [75]). Neither the Vineland-3 Social Interaction and Communication domains nor the ABAS-3 Social and Conceptual domains cover a modern conceptualization of core social communication/interaction dimensions identified in the factor-analytic literature (see review above). Using this literature, Figure 1 displays a review of autism and SCI symptom content for the Vineland-3, ABAS-3, and selected autism symptom measures. As can be seen, the Vineland-3 and ABAS-3 show only item (not scale) coverage of the key SCI sub-dimensions.

#### 2.2.3. Reliability

With the caveat of limited evidence for structural validity and its impact on scoring, the broad domain and sub-domain (skill area) scores show very good scale reliability for the Vineland-3 ([10]; [63]) and ABAS-3 ([30]). Conditional reliability via Rasch or item response theory modeling has not been reported in the literature for either measure. Short-term test–retest reproducibility coefficients (12 to 35 days for the Vineland-3; 5 days to 7 weeks for the ABAS-3) are very good for both measures (r > 0.60), but neither measure has reported longer-term stability coefficients (3+ months) ([2]; [43]). Inter-rater reliability has been found to be good to excellent within forms across interviewers/raters and adequate across different respondents to the same form for the Vineland-3. For the ABAS-3, inter-rater reliability was reported in the manual as good to excellent for raters across forms (in overlapping age bands) as well as good across adult self and informant report versions.

#### 2.2.4. Validity

Both instruments have very good global SCI coverage but, as noted above, the coverage and representation of the more distinct SCI subdomains are limited. For example, the Vineland-3 has many items that assess peer interactions and relationships as well as basic social communication skills ([63]). Coverage of social motivation and perspective-taking is also present but less thorough. The ABAS-3 also has very good but not comprehensive coverage. In particular, the conceptual communication subdomain and the social domain of the ABAS-3 have very good coverage of basic social communication skills and relationship skills and some perspective-taking skills but weaker coverage of social motivation ([30]). Without explicit measurement of this domain, in cases with lower social motivation, initial and ongoing treatment planning could miss the need to focus on increasing affiliative behaviors related to social motivation that are often required to facilitate relationship development. Further, any skill acquisition observed in relationship-building may be challenging to generalize without the child naturally engaging in additional learning opportunities.

Concurrent validity, including the convergent and discriminant validity of the domains and content or skill areas that are measured, is very strong for both instruments ([14]), with evidence across multiple studies for the Vineland-3 ([11]; [66]). Correlations with autism symptom measures have been observed but have tended to be medium in size and differ across studies and measures ([66]; [78]). The Vineland-3 has evidence of domain score sensitivity to change from observational studies and randomized controlled trials ([13]; [53]).

#### 2.2.5. Norming

Both instruments have strong normative representation, with samples approximating US population characteristics at the time of collection. Both tools provide age-adjusted standard scores using very large normative samples. The Vineland-3 includes methods that mix a traditional norming approach and accounts for distribution skewness and non-linear age trends. The ABAS-3 uses traditional normative methods, including age bins and matching to the US census. Neither measure uses a modern continuous norming approach ([36]). Continuous norms are a relatively recent innovation with great applicability to developmental assessments. In this approach, norms are built using prediction models (also called regression-based norming) and the models can simultaneously account for multiple demographic factors and their interactions. Our group has recently shown that a particular form of continuous norming, generalized additive modeling, can result in sensitive normative adjustments with modest neurotypical sample sizes, more accurately accounting for non-linear developmental trajectories and interactions between demographic factors (age by sex) than simpler norming methods ([23]) (Appendix A).

#### 2.2.6. Scoring, Reporting, and Change Measurement

The Vineland-3 provides raw and norm-referenced (age-adjusted) scores for interpretation. The Vineland-3 also provides growth scores for change measurement. Reliable change scores are not available for either instrument. Meaningful change score levels have been recently published for the Vineland-3 ([5]). Depending on the metric, the anchor-based and distribution-based meaningful change scores tend to be quite small and at or below the standard error of measurement for composite or domain scores. Thus, the very minimal change required for meaningful improvement on the Vineland-3 suggests that most reliable changes are likely to also be viewed as meaningful.

### 2.3. Pragmatic Considerations

Table 3 presents pragmatic criteria relevant to outcome assessment in real-world CIBI settings, with an overview of the instruments relevant to each characteristic. The Vineland-3 and ABAS-3 offer automated online administration and scoring with the visual display of measure results and interpretive statements to facilitate cross-sectional use. Both measures also provide behavior lists that can inform target identification, although, as noted above, in the content validity section, coverage of relevant behaviors is not as comprehensive as what would be available through a skills-based assessment ([47]; [65]). Limited intervention strategy guidance is also available, predominantly around the content areas to be addressed. Automated progress monitoring is available in the online system but is limited in terms of tracking longitudinal progress from multiple repeated assessments. The connection of target behavior identification with intervention planning and medical necessity processes is not available. Neither measure offers the ability to aggregate results with other measures that are important for a thorough CIBI outcome assessment to facilitate the clinical workflow.

## 3. Discussion

### 3.1. Vineland-3 and ABAS-3 Strengths

The Vineland-3 and ABAS-3 are standardized, norm-referenced assessments with a long history of clinical and research use relevant to autism and other developmental disabilities. The instrument scores have strong scale reliability, test–retest reproducibility, and inter-rater reliability across forms and raters. They also have strong evidence of construct validity for the broad SCI domain and good potential to sensitively detect change (particularly for the Vineland-3). Age-adjusted and growth scale value scores (Vineland-3 only) necessary for intervention applications are available along with basic administration and scoring automation and clinical interpretive guidance. This makes them good choices for cross-sectional assessments, particularly for outpatient mental health settings where the assessment of adaptive behavior for intellectual disability determination is a common clinical question. The Vineland-3 and ABAS-3 also may have value for intervention monitoring in situations where longer SCI assessments are feasible and only measurement of the broad SCI domain is required ([35]), with Vineland-3 domain scores having demonstrated evidence of sensitivity to change.

### 3.2. Vineland-3 and ABAS-3 Weaknesses for the CIBI Context

Relative to their long history of use and strong psychometric properties for cross-sectional use in outpatient mental health contexts, the Vineland-3 and ABAS-3 were not designed for a periodic CIBI outcome assessment process, where measurement of specific SCI dimensions is needed. While other standardized, norm-referenced assessments are sometimes co-administered to compensate for these deficiencies, this adds a significant burden to the process. The Vineland-3 and ABAS-3 also do not include many of the modern measure development procedures relevant to building assessments for a specific purpose or population, such as parent/caregiver and patient/client stakeholder inclusion throughout the process. Unfortunately, in spite of extensive use over many years, the instruments have undergone only partial psychometric evaluations. More specifically, both instruments have received very limited evaluation in terms of their item-level factor structure. When the structure was evaluated using scale scores, originally proposed factor solutions tended to show either poor or adequate rather than excellent fit to support scoring, and importantly, for the Vineland-3, there was also evidence for an alternative structure. In addition, for both instruments, there is only limited evidence of measurement consistency across relevant demographic or clinical subgroups, with the exception of some evidence of subscale level measurement invariance across demographic groups for the Vineland-3. Both measures use Likert frequency scales that are not well-connected to how progress is evaluated in CIBI. Specifically, frequency scaling leaves mid-points of “Sometimes” or “Sometimes” and “Often” that are not linked to the level of prompting or independence that the individual is able to achieve. This makes using item-level scores to inform programmatic decisions unclear. Additionally, content coverage is not sufficient as a stand-alone tool for intervention targeting and goal/objective selection in CIBI and the length of these measures makes supplementation with longer tools less attractive.

The presence of growth value scores (person ability estimates) is a potentially attractive feature of these instruments, but the fact that these scores are only available for broad domain structure makes them less useful when conducting CIBI follow-up evaluations to inform future intervention planning. For example, not having growth scores for specific SCI dimensions makes it difficult to discern if one should continue targeting social motivation and basic social communication skills or move to higher-order cognitive skills. Thus, even though the Vineland-3 domain scores have demonstrated sensitivity to change in several observational and clinical trial contexts, the lack of specific SCI dimension growth scoring makes it unclear if a broad socialization domain sensitivity to change translates to specific SCI dimensions. Finally, and crucially from the standpoint of usability and implementation in clinical practice, although both instruments have some basic automation and interpretive guidance that facilitates their use in cross-sectional assessment settings, limited functionality is available to increase the efficiency of CIBI progress monitoring. As a result, instrument deployment and the possibility of seamlessly integrating these instruments with other assessments are limited.

### 3.3. Future Directions

The above-described weaknesses in the most commonly used adaptive functioning measures emphasize the urgent need to rethink how SCI, and more broadly adaptive behavior, is evaluated and considered as part of a larger set of domains/constructs essential for CIBI baseline and follow-up evaluations. Figure 2 presents a list of domains, including SCI, which may be useful to assess for many ASD cases receiving CIBI to improve the treatment process and provide a more in-depth accounting of intervention benefits. The described domains were identified through an iterative process encompassing a systematic review of the literature and input from several stakeholder groups and were part of a broader process aimed at identifying key domains of functioning in ASD and related neurogenetic syndromes, with relevance to appropriate outcome assessment ([34]). While not all constructs (and associated measurement tools) would be required for all individuals receiving CIBI, it is clear that the current adaptive function measures assess only a subset of relevant SCI behaviors and only a subset of other adaptive behavior constructs that are not aligned with the current understanding of the comprehensive structure and multi-faceted nature of the broad adaptive functioning construct. Simply adding one or more of the existing autism-specific symptom measures leaves many relevant constructs unaccounted for.

#### 3.3.1. Combining Norm-Referenced and Skills-Based Assessments

Many CIBI practitioners, including nearly all BCBAs, are trained in skills-based or criterion-referenced assessment. Although skills and criterion-referenced methods lack norm-referencing and hence the ability to identify intervention targets specifically tied to behavioral domains deviating from normative expectation, they could be used after norm-referenced assessments to identify specific behaviors within behavioral domains that cannot be independently demonstrated by the patient. Thus, a more efficient process would be to deploy a battery of instruments that includes briefer norm-referenced assessments with comprehensive coverage of the relevant domains and a skills-based instrument. This combined norm-referenced and skills-based or criterion-referenced battery would be able to rapidly identify deviations from normative expectation (domains to target) and inform the administration and interpretation of the skills-based assessment, and the overlap between domains identified via norm-referenced assessments and skills-based assessment results could be used to generate plausible intervention target lists. The battery and approach described above leverages one of the strengths of norm-referenced assessments—the ability to identify deviation from developmental expectation ([1])—and ensures the comprehensive coverage and direct observation of functional behaviors via skills-based or criterion-referenced assessments (Appendix A). For contexts where very detailed assessment is desired, it would also be possible to supplement the above strategy with additional measures and checklists of daily living skills and functional needs (e.g., Essentials for Living, the Early Start Denver Model checklist, etc.).

Identifying a briefer battery of standardized, norm-referenced instruments with good psychometric properties, including sufficient score range and conditional reliability across the domain trait, facilitates repeated deployment of the instruments for the temporally sensitive monitoring of CIBI progress. This broad measurement approach to derive a comprehensive, psychometrically robust, yet brief battery can be effective in tracking the full range of intervention responses, including those occurring in targeted and non-targeted domains. It may seem counter-intuitive to track non-target domains, but CIBI is an intensive and comprehensive strategy and, therefore, is very likely to have “off-target” effects on a range of behaviors, including attention, anxiety, sleep, the child’s and family’s quality of life, etc. Thus, it is essential that future practice recommendations and payor policies for CIBI outcome assessment ensure the adequate capture of potentially relevant behavioral domains. The inclusion of a broad monitoring approach with target and non-target domains may also be useful for informing the intervention strategy. For example, even if family and caregiver quality of life is high at baseline evaluation, a future reduction in this domain could signal that parent-mediated intervention is exacerbating stress and potentially reducing the value of the broader treatment strategy. In these scenarios, the clinician may decide to recommend a temporary reduction in parent-mediated goals to reduce stress. Thus, deploying a set of briefer instruments permits more thorough domain coverage and a fuller accounting of CIBI value while also providing information that could signal the need for changes in clinical management. This measurement approach is also likely to provide more value for ongoing clinical quality improvement and value-based care contracting.

Recently, there has been substantial progress in developing and evaluating norm-referenced assessments that assess domains relevant to CIBI outcome assessment, including instruments that are briefer, with strong psychometric properties and the broader coverage of a range of domains and subdomains relevant to CIBI outcome evaluation ([12]; [15]; [20], [21]; [28]; [42]; [48]; [50]; [68], [73], [69]; [77]).

Adopting these newly developed instruments, created using best practices and stakeholder inclusion specifically for an autism intervention context, would facilitate the measurement of a wide array of CIBI-relevant neurobehavioral domains. Coupling these measures with an existing skill-based assessment (e.g., VB-MAPP, ABBLS-R, AFLS, PEAK) offers several strengths. First, the inclusion of brief norm-referenced measures of key domains would permit the identification of the patient’s level of functioning in each of the SCI dimensions (as well as other relevant domains) and tracking this level over time with less burden on raters than batteries with longer assessments. Second, the inclusion of the skills-based or criterion-referenced assessment permits the direct collection of detailed information on key adaptive behaviors, including SCI-relevant behaviors, to supplement norm-referenced data and facilitate the careful selection of intervention targets. While skills-based assessments have significant psychometric limitations ([45]), coupling them with brief but psychometrically rigorous norm-referenced assessments covers both major tasks relevant to CIBI outcome assessment—identifying and tracking key behavior domains and choosing intervention targets. Finally, the combination of informant report norm-referenced scales with skills-based or criterion-referenced assessments provides multi-modal assessment (parent/caregiver report, clinician or educator rating, and the direct observation of relevant behaviors). Multi-modal assessment yields a fuller picture of child behavior and provides more data on the therapeutic strategy, setting, and parent/caregiver training that would be difficult to attain with a single modality. The major potential weakness of this approach is that the combination results in a longer and more burdensome assessment process than a single instrument or modality alone. However, this tradeoff is likely acceptable given the need for broad strong coverage of SCI behaviors as well as other adaptive behaviors and child and family functioning.

#### 3.3.2. Adopting the ICHOM Battery

A second option would be to adopt the ICHOM autism spectrum disorder standard set battery (Track A with commercial tools or Track B with a mixture of commercial and free or low-cost tools) (https://www.ichom.org/patient-centered-outcome-measure/autism-spectrum-disorder/ (accessed on 1 August 2024)) ([34]). This option is attractive for several reasons. First, the ICHOM battery was developed with stakeholder inclusion and explicit consideration of the domains of assessment that would be most valuable for an autism intervention context, including core autism symptoms, SCI behavior, family functioning, and quality of life. The process also reviewed and prioritized tools based on available psychometric information and used a modified Delphi process to develop agreement on recommendations. The major potential weaknesses of this approach are that (1) it recapitulates criticisms of the Vineland-3 coverage of specific SCI dimensions, as the Social Responsiveness Scale-Second Edition (SRS-2) does not adequately compensate for these limitations, and (2) the lack of inclusion of skills-based or criterion-referenced assessment results in data based largely on a single source (e.g., parent/caregiver) and provides limited data for SCI intervention target selection. Furthermore, many of the instruments included in the ICHOM battery lack the pragmatic considerations described above (e.g., no automated administration, scoring, or reporting).

#### 3.3.3. Revision of Vineland-3 and ABAS-3 for CIBI

A third option would be to revise the existing adaptive behavior instruments for a CIBI outcome assessment context. Given the above-reviewed limited construct coverage of current adaptive behavior instruments, this approach would require augmenting existing item content to ensure the sufficient coverage of SCI subdimensions to support scoring as well as the potential removal of existing items/subscales (e.g., the Vineland-3 written expression subscale). The revised instruments would need to undergo significant psychometric evaluation before implementation, with particular emphasis on developing a replicable factor structure and measurement invariance across key demographic and clinical characteristics. These revisions would require significant time and resources and this option would also need to consider either (i) supplementing revised adaptive function measures with instruments that evaluate other relevant domains, such as the Behavior Assessment System for Children-Third Edition for other psychopathology domains, such as the WHOQoL-BREF ([42]) or the QI-Disability ([12]) to evaluate quality of life/flourishing, the Behavior Rating Inventory of Executive Function (BRIEF-2) for executive functioning ([28]), and a measure of core autism symptoms; (ii) or developing additional item sets that would be incorporated into the revised adaptive assessments to assess these additional content areas. Similar to the ICHOM battery, this approach would also focus on informant report without direct collection of key SCI behaviors. Thus, while this option may have strong long-term potential if the necessary revision processes are undertaken, it is not likely to be an adequate short-term solution. Finally, this approach would also require addressing the current practical limitations of the Vineland-3 and ABAS-3, such as longitudinal score monitoring, linking results to potential intervention targets, and aiding the clinician in detecting reliable change to inform clinical management decisions.

## 4. Conclusions

In contrast to their strong utility in cross-sectional outpatient mental health assessment settings, the Vineland-3 and ABAS-3 present with a range of psychometric and conceptual limitations that significantly impact their utility as outcome assessments. These legacy adaptive function measures have long administration times; lack an in-depth coverage of key aspects of adaptive functioning that are aligned with the current theoretical models and empirical factorial evidence and findings from the studies that have examined the outcome priorities of the autistic community and their families; have inadequate psychometric information to support their scoring, interpretation, and monitoring; and provide only a subset of the practical implementation features that would facilitate use in everyday CIBI practice. Providers and payors should reconsider the adoption of these measures as part of a CIBI outcome assessment battery. Instead, a more optimized CIBI assessment approach is recommended that collects multi-modal information and leverages the strengths of newly developed norm-referenced assessments for the autism behavioral intervention context coupled with skills-based and neurodiversity-affirming assessments that provide deep coverage of key social and functional domains relevant to choosing specific intervention targets. In addition, once such assessments are developed and stringently validated, it will be essential to explore and establish their cross-cultural validity and, if needed, make appropriate updates and refinements.

## Figures and Tables

**Figure 1 behavsci-15-00722-f001:**
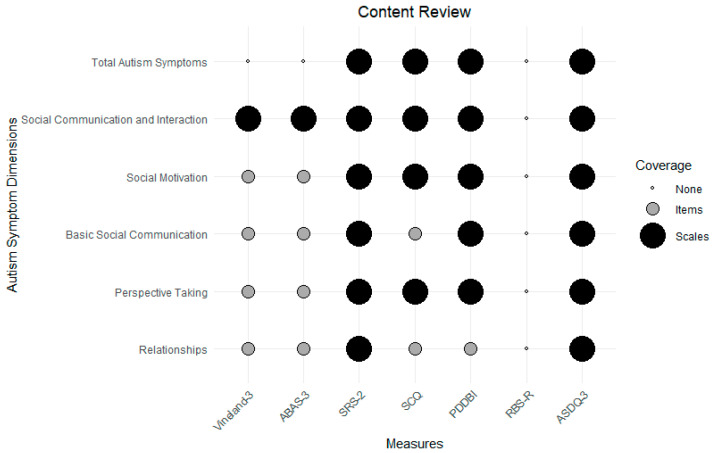
A content review of adaptive function and selected autism symptom measures for total autism symptoms, the SCI domain, and specific SCI dimensions identified in the factor-analytic literature ([6]; [22]; [48]; [59]; [70]). Note, SRS-2 = Social Responsiveness Scale-Second Edition, SCQ = Social Communication Questionnaire, PDDBI = Pervasive Developmental Disorder Behavior Inventory, RBS-R = Repetitive Behavior Scale-Revised, ASDQ-3 = Autism Symptom Dimensions Questionnaire-Version 3.0.

**Figure 2 behavsci-15-00722-f002:**
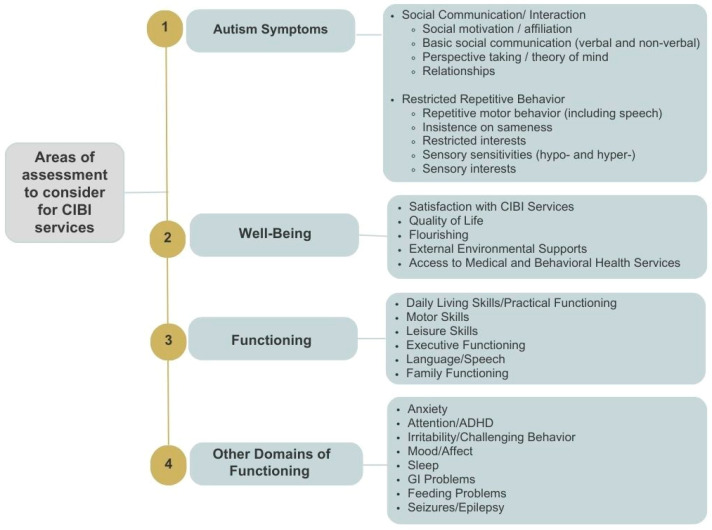
Domains of coverage for initial and periodic CIBI outcome assessment, derived from prior processes for eliciting input from caregiver/patient and clinician/scientist ASD stakeholders. Several prior large-sample studies support the Social Communication/Interaction framework detailed above ([6]; [22]; [48]; [59]; [70]). Note, cognition, including general cognitive ability and verbal and nonverbal aspects of intelligence, is a frequent outcome measure in research studies but would be very difficult to include in an initial and periodic CIBI outcome assessment process for everyday practice due to the additional time and expertise required for cognitive test administration and the additional cost added to the process. Unfortunately, as noted above, because both the Vineland-3 and ABAS-3 include a large but not comprehensive list of SCIs, adding skills-based assessments to compensate for incomplete coverage is often not practically feasible. This approach would lead to very long and burdensome assessments without adequately covering all of the relevant domains that are most essential for treatment planning. Importantly, the Vineland-3 and ABAS-3 did not systematically engage with the relevant stakeholders, in particular with autistic individuals, during the initial development phase. In addition to the increased awareness of the need for an inclusive approach to research design, a number of different gold-standard measurement development and validation frameworks, including the PROMIS (Patient-Reported Outcome Measurement Information System) framework, have emphasized the need to engage the relevant stakeholders in order to ensure not only that the measures do not miss any behaviors that are important to capture, but to also facilitate the acceptability of the instrument. Thus, we review three alternative approaches to improving CIBI outcome assessment: combining skills-based assessments with newly developed standardized, norm-referenced assessments; adopting the International Center for Health Outcome Measurement (ICHOM) ASD standard set ([34]); and developing revised versions of the existing adaptive behavior measures while coupling them with measures of additional domains relevant to CIBI outcome assessment.

**Table 1 behavsci-15-00722-t001:** Measure development guidelines most relevant to the CIBI outcome assessment process and the evaluation of the Vineland-3 and ABAS-3 across these guidelines.

		Vineland-3	ABAS-3
**1**	Stakeholder involvement from conceptualization to validation	Partial—prior administrators, clinicians, and clinical researchers	Partial—experts, literature review with item refinement including stakeholders
**2**	Use of qualitative research processes, including concept elicitation interviews, in determining domains/sub-domains guiding measurement coverage	Not present	Not Present
**3**	Creation of a domain map or disease model to guide measure creation	Not present	Present for IDD, not specific to ABA
**4**	Consideration of CIBI population-specific issues	Partial—IDD population	Partial—IDD population
**5**	Attention to the nature of anticipated measure utilization (e.g., diagnostic or outcome measurement)	Not present	Not present
**6**	Use of cognitive interviewing to evaluate item appropriateness	Partial—for teacher-reported versions	Unknown
**7**	Assessment of item relevance, clarity/readability, and potential bias	Present—review for bias and relevance/importance	Present—review for clarity/relevance

**Table 2 behavsci-15-00722-t002:** Psychometric criteria for evaluation of Vineland-3 and ABAS-3 use in CIBI outcome assessment.

	Criterion	Vineland-3	ABAS-3
* Structure *			
1	Factor structure	Inconsistent findings for subdomains across publications, no item-level analysis	Inconsistent findings, no item-level analysis
2	Measurement invariance/differential item or scale functioning	Inconsistent evidence for clinical IDD groups but good invariance evidence for sex, race/ethnicity, and SES	Invariance evidence for age and sex/gender
3	Measurement model guides scoring	Scoring based on theoretical, not empirical considerations	Scoring based on theoretical, not empirical considerations
* Reliability *			
4	Scale reliability	Very strong for scored scales	Very strong for scored scales
5	Conditional reliability	Not evaluated	Not evaluated
6	Test–retest reproducibility	Very good	Very good
7	Test–retest stability	Not evaluated	Not evaluated
8	Inter-rater reliability	Good to excellent for raters within forms, across interviewers	Good to excellent for raters across forms with overlapping ages, good across adult self and adult informant reports
* Validity *			
9	Content coverage	Good to excellent for broad SCI domain, weaker for specific SCI content areas	Good to excellent for broad SCI domain, weaker for specific SCI content areas
10	Construct coverage	Limited evidence to support domain sub-dimensions, no coverage of key SCI subdomains	Limited evidence to support domain sub-dimensions, no coverage of key SCI subdomains
11	Convergent validity	Strong evidence for broad domain scores across numerous studies, including criterion-related validity in ASD samples; small to medium correlations between Vineland domains/sub-domains and ASD instrument scores; correlations with ASD symptom dimensions are similar across communication, daily living, and social domains	Good evidence for broad domain scores across several studies, including criterion-related validity in ASD samples, correlations with ASD symptom measures are moderate and comparable in size across the three SCQ domains for SCQ; some evidence of convergent validity with ADOS-2
12	Discriminant validity	Excellent	Excellent
13	Sensitivity to change	Evidence for significant and clinically meaningful gains across observational and randomized studies for socialization domain score	Not evaluated
* Norming *			
14	Sample representativeness	Strong representativeness	Strong representativeness
15	Appropriate demographic adjustment	Yes, age adjustment for standard scores	Yes, age adjustment for standard scores
16	Traditional vs. continuous norming	Mixed norming, with accounting for distribution skewness and non-linear age trends in v-scale scoring	Traditional norming, but with a very large sample ~4000 and matching to US census
* Scoring/Reporting *			
17	Validity indicators	Not present	Not present
18	Raw scores	Present by domain and subscale	Present by domain and skill area
19	Norm-referenced scores	Present by domain and subscale	Present by domain and skill area
* Change Measurement *			
20	Growth scores	Present by domain	Not present
21	Reliable change scores	Not present	Not present
22	Clinically meaningful change	Published levels available	Not present

Note. SCQ = Social Communication Questionnaire. ADOS-2 = Autism Diagnostic Observation Schedule-Second Edition.

**Table 3 behavsci-15-00722-t003:** Pragmatic criteria for evaluation of Vineland-3 and ABAS-3 use in real-world CIBI clinical settings.

		Vineland-3	ABAS-3
**1**	Automated online administration	Yes	Yes
**2**	Automated scoring	Yes	Yes
**3**	Visual display of measure results	Yes	Yes
**4**	Automated interpretative statements	Yes	Yes
**5**	Automated connection of results with intervention target identification	Partial—item/target behavior lists	Partial—item/target behavior lists
**6**	Automated clinical guidance for additional assessment, intervention strategy, and referral	Limited intervention strategy guidance	Limited intervention strategy guidance
**7**	Automated progress monitoring	Limited	Limited
**8**	Connection of target identification with intervention planning	Not present	Not present
**9**	Aggregation of other measures and reporting to facilitate clinical workflow	Not present	Not present

Note. CIBI = Comprehensive Intensive Behavioral Intervention. Vineland-3 = Vineland Adaptive Behavior Scales-Third Edition. ABAS-3 = Adaptive Behavior Assessment System-Third Edition.

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
