# Peer review of "A Critical Appraisal of the Measurement of Adaptive Social Communication Behaviors in the Behavioral Intervention Context"

_behavsci, 2025, doi:10.3390/bs15060722_

Round 1
Reviewer 1 Report
Comments and Suggestions for Authors
This article addresses an important aspect of how evidence based assessments might be in tension with treatment planning. Overall, the focus of the paper has been identified but the way it is written needs to be significantly structured and refined in front matter content (introduction/purpose). Back matter content appears to be well organized and written.
- The crux of the paper is not thoroughly or clearly written out until Page 6 line 282-297 – this needs to be put in the abstract and in the beginning of the paper.
- The “why” of the ABAS-3 & Vineland-3 being reviewed needs to be pointed out early on in the paper and the problems that have aroused because of the continued use of the assessment needs to be clarified.
- For example why has it been “so bad” that Vineland & ABAS used?
- What are the negative effects that it has been used clinically in CIBI?
- Has CIBI been using any assessments to do not cause a tension w/treatment planning?
These prior mentioned questions are relevant to the purpose of this paper but remain unanswered.
- Appendix includes a significant amount content – is all this content relevant to the purpose of this paper?
Author Response
Reviewer 1
This article addresses an important aspect of how evidence-based assessments might be in tension with treatment planning. Overall, the focus of the paper has been identified but the way it is written needs to be significantly structured and refined in front matter content (introduction/purpose). Back matter content appears to be well organized and written.
Comment 1: The crux of the paper is not thoroughly or clearly written out until Page 6 line 282-297 – this needs to be put in the abstract and in the beginning of the paper.
Response 1: We have now expanded the opening of the abstract, which now reads as:
“Despite encouraging evidence for the efficacy of comprehensive and intensive behavioral intervention (CIBI) programs, the majority of studies have focused on relatively narrow, deficit-focused outcomes. More specifically, although adaptive social communication and interaction (SCI) are essential for facilitative functioning, the majority of studies have utilized instruments that capture only the severity of SCI symptoms or conflate symptoms and skills within a single score. Thus, given the importance of the comprehensive and appropriate characterization of distinct SCI adaptive skills in CIBI, in this critical review, we focused on providing a critical appraisal of two of the most commonly used adaptive functioning measures—the Vineland Adaptive Behavior Scales – Third Edition (Vineland-3) and the Adaptive Behavior Assessment System – Third Edition (ABAS-3), for characterizing SCI in the behavioral intervention context.”
We have also significantly expanded the opening paragraph by adding the following statements:
“Further, over the last decade, both theoretical frameworks and dimensional initiatives such as the National Institute of Mental Health’s Research Criteria, as well as latent variable modelling studies, have emphasized the multi-dimensional nature of specific aspects of adaptive functioning, in particular of social communication and interaction domain, demonstrating that different subdomains have distinct trajectories and underpinning mechanisms and might thus respond differently to specific treatments. Therefore, by providing only overly broad scores that conflate distinct subdomains and/or skills and symptoms, outcome measures can obscure the potential positive effects of specific treatments. Given these recent developments, it is essential to ensure that outcome measures are psychometrically robust and provide a comprehensive capture of all the relevant adaptive subdomains without conflating distinct constructs. Thus, in this review, we focused on appraising the utility of the most widely used adaptive functioning instruments as outcome measures for behavioral treatments.”
Comment 2: The “why” of the ABAS-3 & Vineland-3 being reviewed needs to be pointed out early on in the paper and the problems that have aroused because of the continued use of the assessment needs to be clarified. For example why has it been “so bad” that Vineland & ABAS used? What are the negative effects that it has been used clinically in CIBI? Has CIBI been using any assessments to do not cause a tension w/treatment planning? These prior mentioned questions are relevant to the purpose of this paper but remain unanswered.
Response 2: Further to the changes in the opening paragraph, we described above, we have provided additional clarifications with regard to the essential properties of the adaptive functioning instruments for their use in the context of outcome measures and highlighted the negative consequences of instruments having poor domain coverage and representation and lack of utilization of modern norming and quantitative means of characterizing treatment-related change at the individual patient level. Described changes read as:
“In addition to the importance of domain coverage and representation, modern psychometric approaches have made significant strides towards regression-based norming and other approaches that can be utilized to optimize instruments to inform treatment selection as well as to quantify treatment-related change at the individual patient level and thus significantly improve the science and practice of behavioral interventions. Given these recent developments, it is essential to ensure that outcome measures are psychometrically robust and provide a comprehensive capture of all the relevant adaptive subdomains without conflating distinct constructs. Further, it is important to establish whether currently used instruments have utilized state-of-the-art psychometrics to provide modern norming, the ability to track change, and facilitate practice through automated online administration, scoring, and interpretation. Thus, in this review, we focused on appraising whether two of the most widely used adaptive functioning instruments—the Vineland Adaptive Behavior Scales – Third Edition (Vineland-3) and the Adaptive Behavior Assessment System – Third Edition (ABAS-3), provide construct coverage in line with the current empirical evidence and incorporate psychometric advances to enable their valid use as outcome measures for behavioral treatments.”
Further to the changes to the opening paragraph, we touch on the described challenges and the consequences at several points throughout the manuscript. For instance, in lines 363 to 386, the manuscript reviews additional limitations of the Vineland-3 and ABAS-3 regarding use in the CIBI context with reference to capturing relevant SCI sub-domains. Lines 412 to 427 mention the limitations of using the interview version of the Vineland-3 in the CIBI context and the lack of evidence to support greater validity for this version.
Comment 3: Appendix includes a significant amount of content – is all this content relevant to the purpose of this paper?
Response 3: We thank the Reviewer for this comment. We wanted to provide as comprehensive overview of the available evidence as possible, given that appendix does not count toward the word limit and does not impact the readability and the flow of the main manuscript, and given that early reviewers of this manuscript requested significant additional information that is relevant but which could distract from the manuscript flow, we decided to keep the Appendix in the current format in case any of the potential readers might wish to seek additional information.
Reviewer 2 Report
Comments and Suggestions for Authors
Thank you very much for inviting me to revise the following manuscript, “A Critical Appraisal of the Measurement of Adaptive Social Communication Behaviours in the Behavioural Intervention Context”.
I have carefully read and revised the manuscript point-to-point. The content is familiar since I supervise the behavioural plans of people with autism.
The authors offer reasoning in both directions:
- ABA functional assessments are not normative but comprehensive
- Adaptive instruments such as VABS and ABAS are normative but not comprehensive in social skill domains.
The manuscript is interesting for BCBAS, but I would include such a manuscript more like a commentary/letter rather than a review.
Introduction
I suggest a new organisation of the introduction since I found it overworded and a bit redundant.
44-50
This section needs details, it is not clear.
67-72
This section needs details, it is not clear.
133-146
It is a bit redundant, you can consider removing unnecessary information and moving it to the appendix (platform, websites, guidelines) throughout the manuscript.
147-173
As specified above, consider reorganising the topic in arguments, such as monitoring interventions and psychometric proprieties.
232-285
This section needs to be ameliorated since it is the theoretical core of your dissertation. Which subdomains would cover the social skills you study? Which the insights for clinicians?
Consider merging the following paragraphs: 1.6, 1.7, and 1.8, removing redundant.
282-297
I noted a certain repetition of previously cited research questions.
I suggest using the appendices to render the manuscript more readable.
Method
2.1 (diverse contents are already previously addressed by the authors)
Table 2 (I suggest adding citations close to the item you declare for each criterion even if you explain them in a further paragraph)
360-371
It is not clear if the aspects that the author states are a basis for the VABS, Please extend and revise in detail. Moreover, you could indicate the pros and cons concerning the online administration.
Why did the authors not consider the cross-cultural aspects of instruments?
General consideration about the method.
The authors highlight the scarcity of item representativity of VABS and ABAS in social domains (Autism related). Nevertheless, I found scarce detail throughout the manuscript, either in theory explanation surrounding an extensive SCI assessment (see from 232 lines) or in a proposal model (the authors discuss such issue solely certain alternatives)
I suppose a tentative is FIGURE 2, but it lacks citations and literature.
Discussion
589-600
This section needs details, it is not clear.
608-616
This section needs details, it is not clear.
What instrument did the authors refer to?
620
Check the line
3.3.2 Adopting the ICHOM battery
This section needs details, it is not clear.
The authors should clarify such a battery in detail and why they exposed it as a model.
The discussion remains solely hypothetical and seems disconnected from the method.
Concluding,
The manuscript provides a critical perspective of functional assessments and adaptive scales in a behavioural context (intervention and monitoring). The aim of the study is interesting.
My concerns remain on the unnecessary information; many sections of the manuscript are not sufficiently treated (concerning the core argument).
Finally, the introduction and discussion need major revisions.
Good luck
Author Response
Reviewer 2
Thank you very much for inviting me to revise the following manuscript, “A Critical Appraisal of the Measurement of Adaptive Social Communication Behaviours in the Behavioural Intervention Context”. I have carefully read and revised the manuscript point-to-point. The content is familiar since I supervise the behavioural plans of people with autism. The authors offer reasoning in both directions:
- ABA functional assessments are not normative but comprehensive
- Adaptive instruments such as VABS and ABAS are normative but not comprehensive in social skill domains.
Comment 1: The manuscript is interesting for BCBAs, but I would include such a manuscript more like a commentary/letter rather than a review. I suggest a new organisation of the introduction since I found it overworded and a bit redundant.
Response 1: Per reviewer 1s suggestion, we have added significantly to the abstract and introduction. To address the present reviewer’s concerns we have also clarified or removed potentially redundant or confusing text throughout, especially around SCI domains and assessment (see also responses below). We hope this this revised version is clearer. We also opted to maintain submission as a review given that we conducted an comprehensive review of the Vineland-3 and ABAS-3 psychometric literature and distill this information so that readers can judge the current state of psychometric evidence.
Comment 2: 44-50 This section needs details, it is not clear.
Response 2: We have now provided more specific detail to this statement, please see below.
“Despite encouraging evidence for the efficacy of noted approaches, in particular with regards to the improvements in the IQ and language abilities, the majority of treatment trials have focused on a relatively narrow range of outcomes, emphasizing the need to develop assessments capturing behaviors and skills that are relevant to a person's ability to function across different daily contexts.”
Comment 3: 67-72 This section needs details, it is not clear.
Response 3: We have clarified the statement that the Reviewer has highlighted. It now reads as:
“For instance, in situations where the need for stimulus and environmental control is high, naturalistic delivery is less crucial or relevant, and/or child-led intervention is less likely to be effective, many ABA practitioners, including Board Certified Behavior Analysts, need to adapt their intervention delivery methods to the specific profile of individual’s characteristics, using aspects of ABA with strong developmental consideration. In addition, it is essential for practitioners to take into account how meaningful and valid specific treatment goals are to the given family and individual (Sulek et al., 2024; Waddington et al., 2024).”
Comment 4: 133-146 It is a bit redundant, you can consider removing unnecessary information and moving it to the appendix (platform, websites, guidelines) throughout the manuscript.
Response 4: We thank the Reviewer for this comment. However, given the Reviewer 1’s concern with the extensive information already present in the Appendix and the need to present information that is relevant to everyday clinical practice, we have decided to retain the information.
Comment 5: 147-173 As specified above, consider reorganising the topic in arguments, such as monitoring interventions and psychometric proprieties.
Response 5: Given Reviewer 1s comments regarding the current structure we have opted to maintain the headings but have attempted to provide text clarifications throughout as topics shift.
Comment 6: 232-285 This section needs to be ameliorated since it is the theoretical core of your dissertation. Which subdomains would cover the social skills you study? Which the insights for clinicians?
Response 6: Following the Reviewer’s comment below, we have now merged sections 1.6, 1.7, and 1.8 and this information is promimently featured in the first paragraph of the revised section 1.6.
Comment 7: Consider merging the following paragraphs: 1.6, 1.7, and 1.8, removing redundant.
Response 7: We have now merged these sections and removed redundancies.
Comment 8: 282-297 I noted a certain repetition of previously cited research questions.
Response 8: We have now revised this to remove the repetition.
Comment 9: I suggest using the appendices to render the manuscript more readable.
Response 9: As noted in a response to one of the previous comments by the Reviewer, given the Reviewer 1’s concern with the extensive information already present in the Appendix, we have decided to retain the information.
Method
Comment 10: 2.1 (diverse contents are already previously addressed by the authors)
Response 10: We have now removed redundant text from section 2.1. See lines 370-377.
Comment 11: Table 2 (I suggest adding citations close to the item you declare for each criterion even if you explain them in a further paragraph)
Response 11: Adding citations directly to Table 2 resulted in a very unwieldy and difficult to read table. For section 2.2.2 through 2.2.6 we have checked each psychometric criterion described to make sure that the appropriate reference is listed at the end of the sentence where the characteristic is first mentioned.
Comment 12: 360-371 It is not clear if the aspects that the author states are a basis for the VABS, Please extend and revise in detail. Moreover, you could indicate the pros and cons concerning the online administration.
Response 12: We have checked that section 2.2.1 provides the correct characterization of the Vineland-3 domains and sub-domains as well as the description of the Likert scaling and versions offered. We did not add additional detail based on other reviewer concerns that the manuscript is long and otherwise very detailed and because additional detail on the Vineland-3 and ABAS-3 are available in the manuals which are cited in the text.
Comment 13: Why did the authors not consider the cross-cultural aspects of instruments?
Response 13: This is an excellent point. We completely agree with the Reviewer about the importance of considering cross-cultural aspects of the instruments reviewed in this paper, and more generally. However, psychometric properties that we have reviewed in the current paper, in particular, good domain coverage and representation and strong and replicable factor structure and pre-requisites for cross-cultural validity to be considered, and given the identified issues, any findings with regard to the cross-cultural validity of the Vineland-3 and ABAS-3 would be difficult to interpret and contextualize. We have now added the following statement at the end of the discussion section to highlight the importance of the cross-cultural validity aspect.
“In addition, once such assessments are developed and stringently validated, it will be essential to explore and establish their cross-cultural validity, and if needed, make appropriate updates and refinements.”
Comment 14: General consideration about the method. The authors highlight the scarcity of item representativity of VABS and ABAS in social domains (Autism related). Nevertheless, I found scarce detail throughout the manuscript, either in theory explanation surrounding an extensive SCI assessment (see from 232 lines) or in a proposal model (the authors discuss such issue solely certain alternatives). I suppose a tentative is FIGURE 2, but it lacks citations and literature.
Response 14: We thank the Reviewer for raising this issue. As Reviewer has noted, we have summarized information with regards to the conceptual and empirical model of the SCI in Figure 2. In addition, we have provided an overview of the relevant literature that has identified specific SCI subdomains in the section 1.6. We have now added references of the relevant factorial literature summarized in section 1.6. to the Figure 2 and Table S2.
Discussion
Comment 15: 589-600 This section needs details, it is not clear.
Response 15: This section has now been revised for clarity. See new lines 640-656.
Comment 16: 608-616 This section needs details, it is not clear. What instrument did the authors refer to?
Response 16: In the section that the Reviewer highlighted here, we didn’t refer to any single instrument, rather we provided a brief overview of the specific measurement features and the potential approach that could be used to derive a comprehensive, psychometrically robust yet brief assessment battery. We have now clarified this.
Comment 17: 620 Check the line
Response 17: We have now modified the statement to improve the readability.
Comment 18: 3.3.2 Adopting the ICHOM battery This section needs details, it is not clear. The authors should clarify such a battery in detail and why they exposed it as a model.
Response 18: We have now included the link to the ICHOM ASD standard set. This allows readers to review the standard set Track A and B as well as the process for developing this set. As previously noted, we have focused on ICHOM battery given the prominence in the clinical field and due to the fact that it was developed with stakeholder inclusion, specifically considered which domains are most valuable to be assessed for an autism intervention context, and importantly, considered psychometric properties of the assessments as a key criteria for deriving specific recommendations.
Comment 19: The discussion remains solely hypothetical and seems disconnected from the method.
Response 19: In the discussion section, we have drawn on the evidence provided in the method to provide specific recommendations with regard to the current state of the CIBI assessments, more specifically, SCI adaptive functioning assessments, as well as essential future steps to ensure necessary improvements to the currently available statement approaches. We hope that the changes made have strengthened this section.
Comment 20: Concluding, the manuscript provides a critical perspective of functional assessments and adaptive scales in a behavioural context (intervention and monitoring). The aim of the study is interesting. My concerns remain on the unnecessary information; many sections of the manuscript are not sufficiently treated (concerning the core argument). Finally, the introduction and discussion need major revisions. Good luck
Response 20: We thank the Reviewer for this positive comment as well as detailed and constructive comments above. We hope that the changes made throughout the manuscript have adequately addressed highlighted issues and strengthened the manuscript.
Reviewer 3 Report
Comments and Suggestions for Authors
I reviewed the manuscript "A Critical Appraisal of the Measurement...." I found it presented a timely and critical evaluation of the use of the Vineland-3 and ABAS-3 adaptive behavior measures in the context of Comprehensive and Intensive Behavioral Intervention for individuals with autism spectrum disorder. The authors provide a well-articulated critique of the developmental origins, psychometric limitations, and practical constraints of the instruments when applied to intervention outcome assessment, particularly in light of modern conceptualizations of social communication and interaction (SCI).
The manuscript is well-written, informative, and provides a compelling rationale for shifting toward more context-specific outcome measurement tools. It also addresses a key translational gap in autism intervention science and offers actionable insights for researchers, clinicians, and policymakers.
I recommend acceptance with minor revisions, contingent upon the authors addressing the specific points below.
Strengths
1. The manuscript addresses an urgent need in ASD services: evaluating the adequacy of adaptive functioning measures for treatment monitoring. This is especially pertinent given the growing demand for accountability in behavioral health outcomes.
2. The psychometric evaluation (factor structure, reliability, validity, and norming) is rigorous, and the supplementary material enhances transparency and replicability.
3. The manuscript aligns outcome assessment practices with a neurodiversity-affirming framework, moving emphasis from a deficit-focused measurement to more holistic, person-centered outcomes.
4. The manuscript effectively argues for future development of optimized instruments for the CIBI setting by identifying a lack of subdomain-specific measurement and contextual misalignment of current tools.
Suggestions for Improvement
1. While Table 1 notes the partial inclusion of stakeholders in the Vineland-3 and ABAS-3 development, the discussion could better underscore the implications of omitting autistic individuals and caregivers from early phases of measure design, especially given the manuscript’s emphasis on social validity and lived experience. The authors may wish to expand on these implications in the “Discussion” section and suggest specific stakeholder-engagement practices for future tool development.
2. The pragmatic limitations of both measures are described well (e.g., lack of integration with intervention planning). However, concrete alternatives or next steps could be further developed.
The authors could suggest specific existing tools, frameworks (e.g., skills-based assessments like AFLS or VB-MAPP), or hybrid approaches for bridging the gap in clinical practice until better tools are available.
3. The argument for SCI subdomain differentiation is strong. Still, it could be more impactful with a brief illustrative case example or a more tangible link to clinical decision-making (if in fits the journal's style). The authors may wish to consider adding a hypothetical vignette or summary example showing how subdomain-level data could alter intervention targets compared to domain-level scores.
Minor Comments and Edits
1. Terminology Clarification: The manuscript uses both “SCI” and “adaptive social communication and interaction behaviors.” For clarity and consistency, consider standardizing the terminology.
2. Tables S1 and S2: These are informative, but they would benefit from cross-referencing in the main text for easier navigation and stronger integration.
3. A few citations in the text are listed as “under review” or “in press.” If these have been accepted or published, please update.
Review Conclusion
I believe the manuscript contributes significantly to the literature on outcome measurement in ASD interventions. It provides a solid critique of current tools and a forward-looking vision for improvement. I recommend acceptance after minor revisions, enhancing clarity, usability, and translational value for clinicians and researchers alike. Please feel free to contact me if you need help integrating any of these suggestions into the manuscript text.
Author Response
Reviewer 3
Comment 1: I reviewed the manuscript "A Critical Appraisal of the Measurement...." I found it presented a timely and critical evaluation of the use of the Vineland-3 and ABAS-3 adaptive behavior measures in the context of Comprehensive and Intensive Behavioral Intervention for individuals with autism spectrum disorder. The authors provide a well-articulated critique of the developmental origins, psychometric limitations, and practical constraints of the instruments when applied to intervention outcome assessment, particularly in light of modern conceptualizations of social communication and interaction (SCI).
The manuscript is well-written, informative, and provides a compelling rationale for shifting toward more context-specific outcome measurement tools. It also addresses a key translational gap in autism intervention science and offers actionable insights for researchers, clinicians, and policymakers.
I recommend acceptance with minor revisions, contingent upon the authors addressing the specific points below.
Strengths
- The manuscript addresses an urgent need in ASD services: evaluating the adequacy of adaptive functioning measures for treatment monitoring. This is especially pertinent given the growing demand for accountability in behavioral health outcomes.
2. The psychometric evaluation (factor structure, reliability, validity, and norming) is rigorous, and the supplementary material enhances transparency and replicability.
3. The manuscript aligns outcome assessment practices with a neurodiversity-affirming framework, moving emphasis from a deficit-focused measurement to more holistic, person-centered outcomes.
4. The manuscript effectively argues for future development of optimized instruments for the CIBI setting by identifying a lack of subdomain-specific measurement and contextual misalignment of current tools.
Response 1: We would like to thank the Reviewer for these positive comments and for thoughtful and constructive suggestions below that we believe have significantly strengthen the quality of our submission.
Suggestions for Improvement
Comment 2: While Table 1 notes the partial inclusion of stakeholders in the Vineland-3 and ABAS-3 development, the discussion could better underscore the implications of omitting autistic individuals and caregivers from early phases of measure design, especially given the manuscript’s emphasis on social validity and lived experience. The authors may wish to expand on these implications in the “Discussion” section and suggest specific stakeholder-engagement practices for future tool development.
Response 2: We thank the Reviewer for raising this issue. We completely agree that this is an important component to discuss and emphasize, thus, we have added the following paragraph in the discussion section:
“Importantly, Vineland-3 and ABAS-3 have not systematically engaged with relevant stakeholders, in particular with autistic individuals, during the initial development phase. In addition to the increased awareness of the need for inclusive approach to the research design, a number of different gold-standard measurement development and validation frameworks, including the PROMIS framework, have emphasized the need to engage the relevant stakeholders in order to ensure not only that measures do not miss any behaviors that are important to capture, but to also facilitate acceptability of the instrument.”
Comment 3: The pragmatic limitations of both measures are described well (e.g., lack of integration with intervention planning). However, concrete alternatives or next steps could be further developed.
The authors could suggest specific existing tools, frameworks (e.g., skills-based assessments like AFLS or VB-MAPP), or hybrid approaches for bridging the gap in clinical practice until better tools are available.
Response 3: Agreed. We have added to section 3.3.3 (lines 750-753) additional pragmatic considerations that would need to be addressed in future revisions of the adaptive measure to improve their use in CIBI outcome assessments.
Comment 4: The argument for SCI subdomain differentiation is strong. Still, it could be more impactful with a brief illustrative case example or a more tangible link to clinical decision-making (if in fits the journal's style). The authors may wish to consider adding a hypothetical vignette or summary example showing how subdomain-level data could alter intervention targets compared to domain-level scores.
Response 4: We thank the Reviewer for this comment. We agree that this would be an excellent way to illustrate the utility, unfortunately, in our understanding, it is not possible to include vignettes in the current format.
Minor Comments and Edits
Comment 5: Terminology Clarification: The manuscript uses both “SCI” and “adaptive social communication and interaction behaviors.” For clarity and consistency, consider standardizing the terminology.
Response 5: We have now made the changes to bring the consistency an used SCI throughout the manuscript.
Comment 6: Tables S1 and S2: These are informative, but they would benefit from cross-referencing in the main text for easier navigation and stronger integration.
Response 6: Callouts have been added to the text for Tables S1 (line 128) and S2 (line 290)
Comment 7: A few citations in the text are listed as “under review” or “in press.” If these have been accepted or published, please update.
Response 7: There have been no updates in terms of the status of cited papers since the original submission.
Comment 8: Review Conclusion
I believe the manuscript contributes significantly to the literature on outcome measurement in ASD interventions. It provides a solid critique of current tools and a forward-looking vision for improvement. I recommend acceptance after minor revisions, enhancing clarity, usability, and translational value for clinicians and researchers alike. Please feel free to contact me if you need help integrating any of these suggestions into the manuscript text.
Response 8: We thank the Reviewer for this and other positive comments and we are very glad that the Reviewer found the paper interesting and potentially impactful.
Round 2
Reviewer 2 Report
Comments and Suggestions for Authors
Thank you for including my suggestions in your manuscript.
I read the paper, listing minor issues:
abstract
line 26
The author refers to a PubMed search without including a search strategy in the method section.
introduction
The authors clarified the research questions, even if their scope surpasses them (describing objectives in several paragraphs of the introduction section).
Generally, the current version is more readable than the previous one.
Nevertheless, similar concepts occur in several sections.
The strengths are clinical reasoning and the range of critical literature.
Method
The authors did not include information regarding the extraction and selection of information.
345
Check the line
If Table 2 collects information from different studies over time, the authors should add the citations close to each result.
432
Check the line
451-458
The authors report only the Vineland sensitivity score (change over time).
Please align these results with the discussions.
2.3/3.2
The authors highlight the lack of clinical assessment (social domains) in standardized interviews. Nevertheless, CIBI researchers gathered clinical information from normative instruments, such as symptoms, adaptive, cognitive, ToM, executive, etc.
Throughout the manuscript, I felt that the raters omitted such clinical aspects involuntarily.
Similarly, the authors discuss a merger of instruments by monitoring the behavioral changes.
Additionally, the authors did not discuss sufficiently the functional assessment scales of CIBI and their association with psychometric developments.
511
aforementioned
651-654
People without training in ABA would not understand such an abbreviation.
Consider the supplemental materials (extension information).
Consider adding Essential for Living and the ESDM checklist to offer an overview to readers.
I hope my suggestions will improve the quality of your informative paper.
Good Luck
Author Response
Reviewer Comment 1: I read the paper, listing minor issues:
abstract
line 26
The author refers to a PubMed search without including a search strategy in the method section.
Response 1: We have now included in the abstract that a PubMed search strategy was included and we have added to the Method section specific details of the search strategy, including years and terms used to identify articles (see lines 349-357 of the methods).
Reviewer Comment 2: introduction
The authors clarified the research questions, even if their scope surpasses them (describing objectives in several paragraphs of the introduction section).
Generally, the current version is more readable than the previous one.
Nevertheless, similar concepts occur in several sections.
The strengths are clinical reasoning and the range of critical literature.
Response 2: Thanks for identifying the modifications we made to the previous draft to reduce redundancy and improve clarity. Based on prior comments from reviewer 1 we had decided to maintain small amounts of redundancy in the introduction section, particularly for less experience readers who might need redundancy to reinforce certain concepts.
Reviewer Comment 3: Method
The authors did not include information regarding the extraction and selection of information.
345
Check the line
Response 3: We now include in the beginning of the Review Methods and Results a description of the search strategy. See reviewed manuscript lines 349-357.
Reviewer Comment 4: If Table 2 collects information from different studies over time, the authors should add the citations close to each result.
432
Check the line
Response 4: We were not able to add the references to Table 2 without making it very unwieldy and difficult to read. In the text where each property of the Vineland and ABAS are described we have now ensured that a reference is listed in close proximity (typically the end of the first sentence). This change can be most clearly seen in sections 2.2.1 (overview and instrument structure and scoring) and 2.2.4 (validity).
Reviewer Comment 5: 451-458
The authors report only the Vineland sensitivity score (change over time).
Please align these results with the discussions.
Response 5: We have now aligned the strength of the Vineland in demonstrating sensitivity to change at the domain score level with the discussion. See lines 563 and 594 where mention has been added of this strength to the discussion.
Reviewer Comment 6: 2.3/3.2
The authors highlight the lack of clinical assessment (social domains) in standardized interviews. Nevertheless, CIBI researchers gathered clinical information from normative instruments, such as symptoms, adaptive, cognitive, ToM, executive, etc.
Throughout the manuscript, I felt that the raters omitted such clinical aspects involuntarily.
Response 6: We have focused on the adaptive function measures that are commonly administered and interpreted in a CIBI context. There are other measures that are sometimes collected, including other norm-referenced measures such as the SRS, SCQ, BRIEF. However, adding these to compensate for the lack of accurate SCI measurement in the existing instruments only adds burden to the assessment process. To be responsive we have added to lines 568-569 that additional standardized norm-referenced assessments are administered with the adaptive measures but also note that this adds burden to the assessment.
Reviewer Comment 7: Similarly, the authors discuss a merger of instruments by monitoring the behavioral changes.
Additionally, the authors did not discuss sufficiently the functional assessment scales of CIBI and their association with psychometric developments.
Response 7: We have opted not to get into functional behavioral assessments in this context because they are focused on a very different aspect of CIBI outcome assessment and are not relevant to every patient with autism. Further the manuscript is already very detailed and adding this dimension might confuse some less experienced readers.
Reviewer Comment 8: 511
aforementioned
Response 8: See above responses.
Reviewer Comment 9: 651-654
People without training in ABA would not understand such an abbreviation.
Consider the supplemental materials (extension information).
Response 9: We have added the abbreviation for the PROMIS (Patient-Reported Outcome Measurement Information System).
Reviewer Comment 10: Consider adding Essential for Living and the ESDM checklist to offer an overview to readers.
Response 10: We have now added mention of these measures to lines 670-673 as a possibility when combining instruments.
Reviewer Comment 11: I hope my suggestions will improve the quality of your informative paper.
Response 11: Thanks we appreciate the careful review and believe the revised manuscript is significantly stronger as a result.